# The Shifted and The Overlooked:
# A Task-oriented Investigation of User-GPT Interactions

**Siru Ouyang[1]**[*]**, Shuohang Wang[2], Yang Liu[2], Ming Zhong[1], Yizhu Jiao[1], Dan Iter[2]**
**Reid Pryzant[2], Chenguang Zhu[2], Heng Ji[1], Jiawei Han[1]**
[1] University of Illinois Urbana-Champaign [2] Microsoft Azure AI
siruo2@illinois.edu

## Abstract

Recent progress in Large Language Models (LLMs) has produced models that exhibit remarkable performance across a variety of NLP tasks. However, it remains unclear whether the existing focus of NLP research accurately captures the genuine requirements of human users. This paper provides a comprehensive analysis of the divergence between current NLP research and the needs of real-world NLP applications via a large-scale collection of user-GPT conversations. We analyze a large-scale collection of real user queries to GPT. We compare these queries against existing NLP benchmark tasks and identify a significant gap between the tasks that users frequently request from LLMs and the tasks that are commonly studied in academic research. For example, we find that tasks such as "design" and "planning" are prevalent in user interactions but are largely neglected or different from traditional NLP benchmarks. We investigate these overlooked tasks, dissect the practical challenges they pose, and provide insights toward a roadmap to make LLMs better aligned with user needs.

## 1 Introduction

Over the past years, the NLP community has witnessed several paradigm shifts in technology followed by renewed research focus on applications that test the limits of this technology (Sun et al., 2022). For example, distributed word representations (Landauer et al., 1998; Mikolov et al., 2013) enabled a better characterization of the semantic similarity between words, entailing NLP research gravitated towards tasks like sentiment analysis and dependency parsing (Klein and Manning, 2003). Subsequent technologies like the transformer architecture (Vaswani et al., 2017) and contextual word representations (Devlin et al., 2019; Peters et al., 2018) further expanded the space of possible applications and the edge of NLP research, such

as machine translation (Bahdanau et al., 2015) and document summarization (Tan et al., 2017).

Most recently, large language models (LLMs) (Brown et al., 2020; Chowdhery et al., 2022) such as ChatGPT, emerged as powerful tools capable of achieving unprecedented success across a broad spectrum of NLP tasks (Jiao et al., 2023; Hendrycks et al., 2021b; Clark et al., 2018). These models have become accessible and popular among non-NLP experts, opening the door for many new user applications.

The flood of new applications and the sharing of user interactions with LLMs (Tay et al., 2023) provide a great opportunity to closely examine the distribution of real applications users need on a daily basis. After a detailed analysis, we identify a conspicuous gap between real-world user queries and established NLP benchmarks, suggesting another shift in NLP focus is needed. To systematically analyze the phenomenon and to bridge the gap, we conduct a battery of experiments aiming to examine the following aspects:

- What is the distribution of real-world user queries in terms of domain and task types, and how do they shift from traditional NLP benchmarks (§ 3)?

- What are the emerging tasks and requirements from real-world user queries that may be overlooked in previous studies (§ 4)?

We start by investigating ShareGPT[1], a large-scale collection of user-GPT conversations in the real world, containing 94,145 split data samples. ShareGPT has been used for training powerful LLMs (Chiang et al., 2023; Xu et al., 2023) and incorporated into new datasets (Zheng et al., 2023; Gudibande et al., 2023), both showing substantial advantages. Specifically, we design an annotation framework where we employ GPT-4 (OpenAI,

---

[*]Work partially done during internship at Microsoft.

[1]https://sharegpt.com/

2023) to generate the related information for every user query that appears in ShareGPT. We subsequently delve into the obtained data and conduct comprehensive analyses to answer the aforementioned questions[2]. We summarize our key findings as follows:

1. Generally, real-world user queries demonstrate a tendency towards more aligned with daily life with enlarging diverse user bases.
2. We discovered several tasks, including providing advice, designing, planning, etc., that are seldom touched and pose new requirements in the era of LLM.
3. We summarized the shifting trends and challenges, providing insights to fill the gap for both stakeholders and users.

## 2 Methodology

In this section, we employ GPT to annotate the topic/domain and task type of every sample in the ShareGPT collection. The whole annotation and post-processing pipeline is illustrated in Figure 1. We use human evaluation to verify the quality of our automatic annotation.

### 2.1 ShareGPT

ShareGPT is a publically available large-scale collection of user-GPT conversation histories[3]. It is based on a Chrome Extension[4] where users can choose to upload their interactions with GPT. The version we used contains $94,145$ split user-GPT conversations and is previously used to train LLMs such as Vicuna (Chiang et al., 2023). Every sample in ShareGPT is a multi-turn conversation with utterances from both users and the GPT engine.

### 2.2 Self-demonstrated annotation

The goal of annotating each user query is twofold: to identify the underlying task a user is intending to perform (task types), and to understand the subject matter or field (domains) the query pertains to. The annotation process is challenging because i) the annotation requires a comprehensive and suitable predefined set of task types and domains/topics, and ii) the annotation should accurately reflect the genuine

---

[2]Code is available at https://github.com/ozyyshr/ShareGPT_investigation.

[3]The collection of ShareGPT already contains 305,000 samples to the date of this submission, and is dynamically expanding. Since all the data are not publically available, we use the portion that could be downloaded from Huggingface.

[4]https://github.com/domeccleston/sharegpt

requirements expressed in user queries. We chose to employ GPT-4 to conduct a self-demonstrated annotation due to its superior precision and coverage. The annotation process consists of three stages: 1) chain-of-thought prompting, 2) demonstration sampling, and 3) demonstration pool expansion.

**Chain-of-thought prompting.** Chain-of-thought (CoT) (Wei et al., 2022) uses intermediate steps for text generation and improves the performance of LLMs (Chen et al., 2023). To augment GPT-4's proficiency in comprehending and interpreting user queries, we craft our instructions in manual-CoT style, compelling the LLM to deliver the requisite information in a methodical, step-by-step manner. Specifically, we first ask LLM to identify the domain or topic related to the user query. Then, the LLM is prompted to generate a concise one-sentence summary for the given user query as a reference. Finally, drawing on the insights obtained thus far, the LLM is capable of devising creative and accurate task types corresponding to the user query. The generated task types are fine-grained and diverse, spanning from email editing to dream analysis. Overall, we obtain $13,783$ task types and $8,392$ domain labels for all samples in ShareGPT.

**Demonstration sampling.** While the CoT prompting can generate reasonable annotation, it is known that in-context demonstrations can further improve LLM output quality (Wang et al., 2022a). Thus, we select examples from CoT outputs to serve as demonstrations in the second stage. We initiate the demonstration pool with 20 samples of different domains or topics and task types. For every sample, we randomly select $k$ demonstrations from the pool and append them to the instruction in the first stage. The input and output format could be found in Figure 1.

**Demonstration pool expansion.** To encourage diversity and avoid potential bias in demonstration selection (Wang et al., 2022b), we gradually expand the demonstration pool. Since we are asking GPT-4 to generate free-form task types, one challenge here is to avoid generating too divergent task types. Therefore, we maintain a dictionary to document the time of appearance for every task type. If a task type appears more than a fixed ratio $\lambda$ among all the current samples, we then add the current sample containing the task type into the demonstration pool. By enforcing such constraints, the generated free-form task types could be better "clustered" for

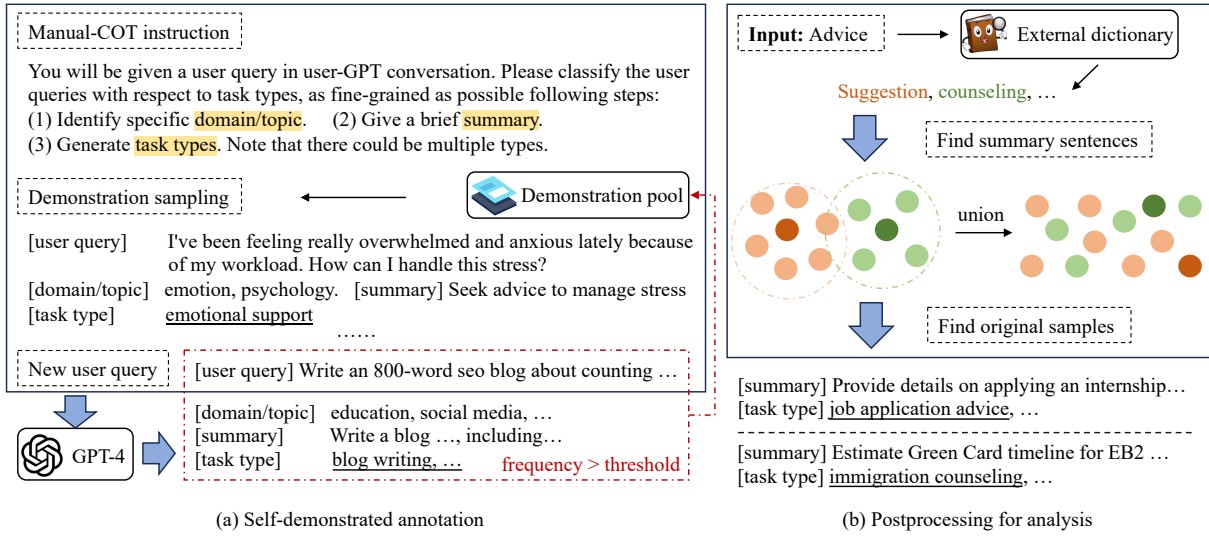

(a) Self-demonstrated annotation

(b) Postprocessing for analysis

Figure 1: General framework of how we conduct step-to-step and self-demonstrated prompting for annotation using GPT-4 (a), and the post-processing for certain words/phrases (b).

further analysis.

**Experiment settings.** We download ShareGPT from Huggingface[5], where the 51k conversations are split into 94k ones due to the length limit for input. For every step, $k$ is set to 3 and $\lambda$ is 0.05. We concatenate 3 samples together and let GPT-4 generate the annotation at once for the balance of speed and quality. To encourage diversity, we set the temperature to 0.4 and it takes around 10 days due to speed limitations in GPT-4 to annotate all the 94k samples in ShareGPT. We plan to release all the annotated results for future related research.

## 2.3 Human Evaluation

To assess the quality of annotation produced by GPT-4, a human evaluation is conducted with a specific focus on the generated free-form task types.

We designed and distributed our human assessment task with Doccano[6]. We recruited 3 graduate students as our human assessors (all of which are paid as research assistants). The assessors all have rich experiences with related NLP and ML research but were not involved in the development of our framework. We randomly selected 100 samples for evaluation. For every sample, we ask the assessors to judge the generated task types in terms of *completeness* and *correctness*. This is to evaluate whether the generated task types are complete and faithful to the original user query. For *completeness*, the scoring scale is 0 (bad) and 1 (good), and

[5] https://huggingface.co/datasets/anon8231489123/ShareGPT_Vicuna_unfiltered
[6] https://github.com/doccano/doccano

Table 1: Human evaluation results in terms of completeness and correctness.

|  | Completeness | Correctness |
| --- | --- | --- |
| Score | 0.95 | 1.76 |
| $\kappa$ | 0.96 | 0.83 |

for *correctness*, the scoring scale is 0 (incorrect), 1 (partly correct) and 2 (correct). The detailed rubric and the interface are shown in Appendix B.

Table 1 demonstrates the results of human evaluation. We can see that with GPT-4 we got reliable annotations for ShareGPT. Notably, none of the 100 samples got "incorrect" annotations. Apart from the scores, we also calculate Fleiss kappa $\kappa$ (Fleiss, 1971) for each metric, both indicating "almost perfect agreement".

## 2.4 Post-processing for analysis

As the domain/topic and task type annotations generated by GPT-4 are free-form words and phrases, clustering the samples poses a significant challenge. For example, "recipe suggestions", "cooking tips" and "cooking advice" all belong to the same task type. To tackle this challenge, we propose a post-processing framework that incorporates three stages shown in Figure 1: (1) a statistical calculation based on heuristic rules, (2) an ensemble with the embedding similarity of summary sentences, and (3) a manual checking process to ensure the best possible quality. Detailed narrations could be found in Appendix A.

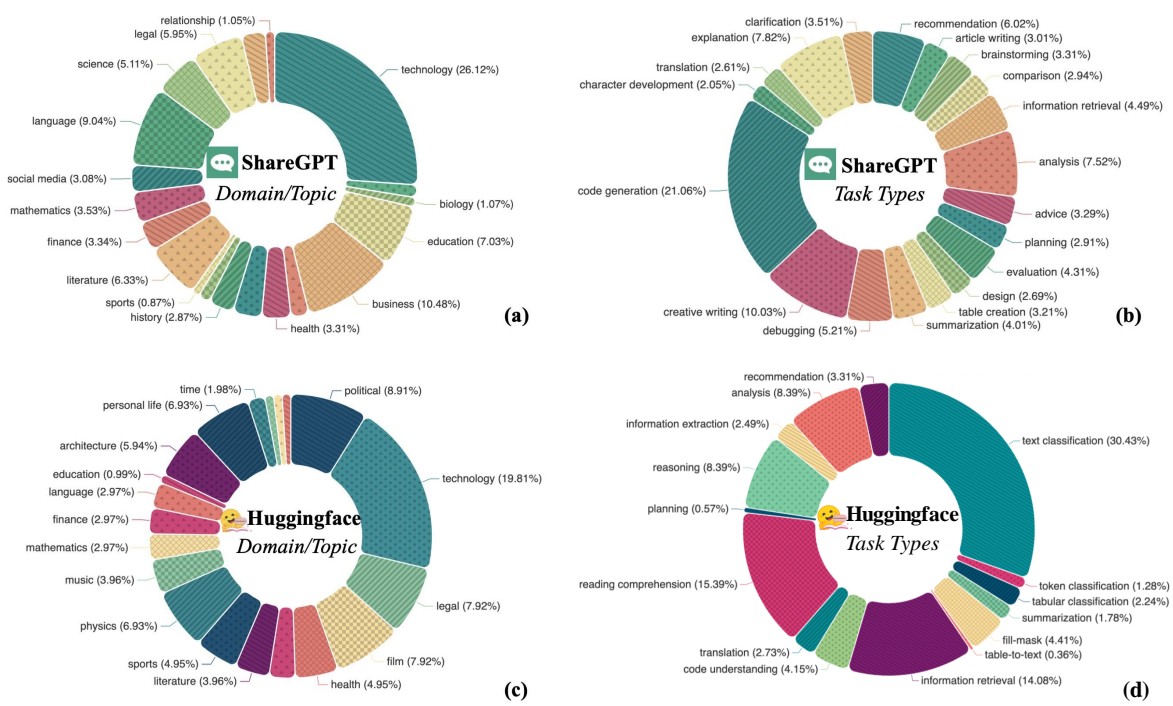

Figure 2: Domain and task types distributions for ShareGPT user query and Huggingface data.

## 3 Overall Investigation

In this section, we first present the overall statistics and analysis of the ShareGPT clustering results. Then we compare the results with conventional NLP datasets.

For conventional NLP datasets, we investigate 2,911 datasets from the Huggingface Datasets[7] collected by Yin et al. (2023). These 2,911 datasets are filtered and selected from the original around 50k datasets in the Huggingface platform with the following conditions: (i) Datasets with no license description or the license does not allow usage. For ethical considerations, the datasets collected are strictly following license restrictions. The number of all datasets fulfilling license requirements is restricted to around 16k. (ii) Datasets that are non-English. Since our investigation focuses on the English language and ShareGPT contains only English samples, datasets collected from Huggingface also follow this paradigm. (iii) Datasets that are related to multi-modal. There is a large portion of datasets in the Huggingface data platform related to multi-modal research. Since we target at NLP community and related topics, we exclude those datasets. (iv) Datasets that are invalid. Some datasets in the Huggingface data platform are empty or cannot be downloaded, which is ignored

by our work.

### 3.1 Domain and Task Distribution

Based on our annotation and clustering results, We plot domain and task type distributions of ShareGPT queries in Figure 2 (a) and (b). "Technology" shares the largest portion of ShareGPT's domain, comprising around a quarter. Other notable domains "education", "business" and "language" make up another quarter. For task types in ShareGPT, around one-fifth is about "code generation", which corresponds to "technology" in domain analysis. "Creative writing" is the second largest portion. The rest of the task types are quite diverse, composing around 2/3 of the whole set.

In the following, we analyze the two mostly seen tasks in ShareGPT, coding and writing assistance, representing 19.9% and 21.3% respectively.

**Coding assistance** Pre-trained models for programming language have been widely explored in the NLP community (Chen et al., 2021; Nijkamp et al., 2023). Current benchmarks used for evaluation (Hendrycks et al., 2021a) are usually in the form of function generation or code completion given certain requirements. More specific task settings include code search (Husain et al., 2019), code translation (Chen et al., 2018), code clone detection (Svajlenko et al., 2014), and code refinement (Tufano et al., 2019). We do observe user

---

[7]https://huggingface.co/datasets?p=0&sort=downloads

queries that are similar to the previously mentioned task settings in ShareGPT, such as code generation (18.6%) and code debugging (9.2%). This reflects that the current coding benchmarks are fitting with real-world scenarios. However, we still notice a non-negligible portion of requests involving higher-level program understanding, such as code simplification and providing design pattern suggestions, which are seldom captured in existing task definitions. We also plot the proportion of the top 10 most frequent programming languages used in Figure 3.

**Writing assistance**   With advancements in the NLP field, writing assistance tools have shown potential beyond grammatical (Ng et al., 2014) and stylistic improvements, now providing aid in content creation and organization. Writing tasks such as story generation (Fan et al., 2018) and style transformation (Shen et al., 2017) are popularly explored in the community. Our analysis of ShareGPT usage confirms this trend. For instance, assistance in article drafting and editing accounts up to 5.1% of the writing assistance requests. Similarly, email editing makes up to 2.6% of the queries. These suggest users rely on AI tools for professional writing communication.

Despite this, we notice a trend of creative writing for a bunch of text formats, spanning from slogans to tutorial writing. Instead of leveraging LLMs to generate everything, we found a noticeable portion of "procedure writing" and "how-to-write" queries, underscoring the importance of explanatory and pedagogical writing aids.

### 3.2   Distribution Difference with Conventional Datasets

To provide a comparison of ShareGPT queries with conventional NLP datasets,we also annotate and cluster the collected 2,911 Huggingface datasets in the same way and present results in Figure 2 (c) and (d). We then interpret the differences from two aspects, domain and task type.

**Domain Shift**   We plot the top 20 most common and representative domains of ShareGPT in Figure 2, which account for around 73% of the entire set. We observe that the domain distribution in ShareGPT is quite diverse. Notably, there is a large portion of technology-related topics. Commonly seen cases involve code writing and debugging. For Huggingface datasets, we randomly select 10 samples from each dataset as an approximation,

and leverage GPT to classify the corresponding domain/topics. Similar to ShareGPT, technology-related domains also share a large portion in Huggingface. However, in Huggingface, political, legal, personal life, and film altogether contribute a large portion. We further look into the data source for Huggingface. We collect the information in meta-data "Dataset Card", where detailed information of the specific dataset is introduced. For example, MRPC dataset (Dolan and Brockett, 2005) has the statement "a corpus of sentence pairs automatically extracted from online **news** sources". We then use GPT to annotate the corresponding data source as an approximation. We find that the majority of datasets are from Wikipedia and news, occupying a portion of over 80%. The rest mainly come from "government reports" and "QA forums". This phenomenon calls for dataset curation specific to domains commonly seen in user queries, and the data source should also incorporate materials in multiple formats apart from Wikipedia and news.

**Task Type Shift**   We categorize different tasks based on metadata statistics for all the datasets from Huggingface, as shown on the right side in Figure 2. We filter out tasks with numbers less than 10, and the rest samples make up to 99.8% of the original set. We observe that question answering and text classification are the top-2 task types, summing up to more than 2/3 of the whole collection. On the contrary, in ShareGPT, almost all the user queries are free-form text generations or generations in a user-required manner.

## 4   Shifted and Overlooked Tasks

In this section, we detail the overlooked tasks discovered in the analysis process of ShareGPT, with concrete examples shown in Table 2. Task selection is based on the distribution in Figure 2 that satisfies two criteria: (1) long-tail tasks of the distribution, summing towards around 40%; and (2) tasks that are not overly skewed in the distribution, constituting around 25% - a comparable figure to the predominant segments. We also summarize the features for each task type in Table 3, along with the potential roadmap. Commonly seen topics and human analysis for performance of LLMs are shown in Figure 6 in Appendix D.

### 4.1   Task of Providing Advice

The task of giving advice occupies a portion of up to 3% in ShareGPT. The concept of a ma-

Table 2: Summary of the long-tail tasks discovered. "‖" indicates concatenated user queries in the same conversation.

| Task type | Characteristic | Proportion | Example |
|---|---|---|---|
| Advice | Offering tailored acvice under given scenarios | 3% | My dad, he's a rude person. he doesn't like me, he makes it clear. he calls me bad names, he has threatened to kick me out. should I be concerned? considering people have suffered through worse ‖ I need money but I can't get it because working at my dad's pub is too much for my autism |
| Design | Construction of some objects or for implementation of an activity | 2.5% | Can you help me think of some logo ideas for "StartUpScout"? ‖ Can you give more similar ideas like "A playful cartoon scout character holding a tech startup icon"? ‖ I liked this idea: A cartoon scout character with binoculars and a backpack, walking towards a futuristic city skyline in the distance. Can you tell me more details? |
| Planning | Providing a sequence of steps to achieve a pre-defined objective. | 2.7% | I'm going on a road trip with my daughter on 3rd April. We will be leaving London and want to be in Nice, France on 9th or 10th April where we'll stay about 3 days. We then need to be in Calais on 15th or 16th to take the Eurotunnel. Everything else is undecided. Can you give a plan for routes where we'll be doing no more than 5 hours driving per day. We have an EV and need to stop for charging. |
| Discussion | Exchanging views or ideas on a specific topic | 3.8% | Should we help local charities or overseas charities? ‖ Some people think that being rich and famous leads to happiness. There is also a saying "Money can buy comfort, but it cannot buy happiness." What do you think? |
| Analysis | Examination of a target for its nature and structure | 7.3% | I need some help analyzing a poem that I've been reading for my English Literature class. The poem is "The Road Not Taken" by Robert Frost. ‖ Thank you, that's a great start. Could you help me analyze how Frost uses metaphor in this poem? |
| Evaluation | Determination of the subject's, properties, based on given rubrics | 4% | I want you act as a resume evaluator. Here are the rules: [Format]: Use a clear, easy-to-read font and suitable layout ... [Content]: Your resume should include... You should be able to read example url and anlyze it. Here are some good example for resume: <https://docs.google.com/...> Each section means 20 point, total 100 points. Just show me explaination and points then organize to table. |

chine offering advice has been present and associated with NLP from as early as the 1970s (Shortliffe et al., 1973) with the development of expert systems (Liao, 2005). At that time, giving advice is usually performed with a consultation program (Scott et al., 1977), which plays the role of an expert consultant in restricted domains, such as health. Since then, the field has evolved significantly, with expanding domains into legal (Pinkwart et al., 2006), finance (Radford, 2003), etc. The advising process back then is more like a QA system, where the user first input background confined by rules, together with a direct question such as "Is Organism-1 a rod or coccus (etc.)?". Later, advice was formally characterized as a multi-step process involving the analysis of the recipient's context and the formulation of a response in natural languages (Saeidi et al., 2018).

Compared with previous advising systems that require rigid rules and were not designed for open-ended dialogue, user queries in ShareGPT are more free-form. Moreover, instead of focusing on restricted professional domains, these requests are more akin to everyday tasks, such as relationship dilemmas and personal careers. This also presents the requests of more general, macro-level guidance, compared with providing micro-specific answers in previous studies.

## 4.2 Task of Designing

Request to design some objects with specifications constitute 2.5% of ShareGPT queries. The task of designing progresses slowly in the field of NLP over the past half-century, and does not have a clear line of development. Previous NLP-related works generally focused on relatively narrow domain-specific topics, such as entity-relationship schema (Habib, 2019) and user interface (Sontakke and Pimpalkar, 2014).

While previous works focus more on the design of structured objects, such as database UI, user queries arising from ShareGPT usually seek designs for more complicated objects, spanning from questionnaires to websites. This indicates a trend that the scope of design has broadened to include not only the structure of tangible objects but also conceptual frameworks and processes. Another significant difference is that design-related queries in ShareGPT ask for more diverse and creative requirements, especially for personalized objects such as icons. Additionally, in ShareGPT, we see instances where users demand designs with very specific constraints, such as adherence to a particular color scheme in interface design. These "visual image design" tasks, though communicating in texts, require the model to have a sense of visual understanding in order to generate aestheti-

cally pleasing outcomes.

## 4.3 Task of Planning

Planning is another important task we identified, which constitutes approximately 2.7% in ShareGPT. Planning in NLP has a long research history. Back in 1969, PLANNER (Hewitt, 1969) was invented as a language for manipulating models in a robot. Follow-up works (Sussman and Winograd, 1970; Bonczek et al., 1979) mostly focus on planning with programming languages for decision-making. Later, a plan was formally defined as an assembly of operators (Grosz and Sidner, 1988) that describes the transition from initial states to final states, where rules and schema were designed for induction. Afterward, planning was employed in different domains and scenarios, such as trajectory (Borrelli et al., 2006), autonomous agents (Chen et al., 2009), and power dispatch (Estevam et al., 2010). Most of the works focus on planning with formal languages under certain rules. Nowadays, many benchmarks related to planning have emerged (Valmeekam et al., 2022; Xie et al., 2023; Wang et al., 2023). Although they require planning with open-form natural languages, they mostly focus on rudimentary tasks with simple actions such as (e.g., "put up a block" for "arrange blocks") (Valmeekam et al., 2023).

The emergence of LLMs has spawned much more free-form and customized formats in planning. One example is illustrated in Table 2, where users ask for planning a route with specific constraints on time, places, and EV charging. We noticed a trend of macro planning, e.g., travel plans, and course plans, instead of planning for micro actions in previous NLP studies. The domains entailed also greatly widen, spreading to many aspects of everyday lives, compared to previous planning systems designed for specific functions/users. Therefore, these planning tasks usually require a higher level ability in personalization, reasoning, and knowledge integration, where follow-up research efforts could lay hands on.

## 4.4 Task of Discussion

Discussion is an interactive and dynamic exchange of ideas or viewpoints, which consists of 3.8% samples in ShareGPT. The act of discussion in NLP is mostly explored in the form of conversation generation with chatbots (Goldenberg, 1992), and they mainly focus on structured discussions in specific domains that require domain expertise, such as political debating (Mirkin et al., 2018). Another notable characteristic is the non-interactivity in many previous works (Zhang et al., 2019; Ouyang et al., 2021), although they promote multi-turn generation (Chen et al., 2020) for several fixed user inputs.

However, user queries in ShareGPT are typically more dynamic and unpredictable, encompassing a vast array of subjects and requiring a deep understanding of various perspectives. For instance, in ShareGPT, there are cases when users initiate philosophical discussions such as "What's the meaning of life?", which may require self-thinking in viewpoints. On the other hand, the discussion process in ShareGPT is quite interactive, which poses challenges in actively taking initiative and even shifting topics. Developing models with certain personalities and empathy to facilitate more effective and meaningful discussions will be helpful.

## 4.5 Task of Analysis

The task of analysis takes up a very large portion of ShareGPT, approximately 7.3%. Textual analysis is a long-standing and crucial branch of NLP. In the early stages, researchers focus on analyzing linguistic features within texts, such as syntax (Floyd, 1963) and discourse (Harris and Harris, 1970). Gradually, they began to wonder "how can analysis of the patterns of words and grammar contribute to an understanding of the meaning" (Stubbs, 1996), accompanied with the investigation in both directions of semantic/contextualized analysis (Mann and Thompson, 1988; Fairclough, 1992) and larger-scale texts. As textual analysis evolves and improves, they are gradually employed in various domains like social science (Fairclough, 2003), medical (Edara et al., 2023), and finance (Fisher et al., 2016) with different objectives like culture (Carley, 1994), sentiment (Nasukawa and Yi, 2003) and opinions (Cambria et al., 2013).

Even though previous research has already covered many fields and objectives, we still observe striking differences when it comes to user queries in ShareGPT. Notably, many previous analysis tasks take the form of classification, e.g., identifying a risk type in a financial report with 25 total risk types as label space (Loughran and McDonald, 2020). User queries, in contrast, usually do not specify the targets/aspects for analysis, as exemplified by the literature analysis case in Table 2. The scope of analysis also varies greatly in ShareGPT, ranging from the overall analysis of classical music

Table 3: Features of the tasks discovered and the potential roadmap.

| Task type | Before LLM | After LLM | Roadmap |
|---|---|---|---|
| Advice | rule-based; limited context analysis; professional domains; micro-specific answers | free-form; open-ended dialogue; general everyday tasks; macro-level guidance | emotion perceivable; personalization; |
| Design | standardized requirement; tangible and structured objects; one-off, static design; | creative design; user-based constraints; expansion to conceptual frameworks and processes; interactive design with feedback; | multi-modality; interactivity; |
| Planning | formal/open-form language; context-insensitive; micro-level actions in specific domains; | free-form language; user-context understanding; macro-level planning for many aspects in daily life; | better reasoning; world-knowledge |
| Discussion | non-interactive with pre-defined inputs; structured around specific domains; | highly interactive, reacting dynamically; encompass broader subjects; personalized with empathy | personalization; interactivity; |
| Analysis | mostly classification; pre-defined aspects; limited domains focused; | free-form input; mostly unspecified targets; wider analysis scope; | multi-modality; better reasoning; |
| Evaluation | mostly plain texts for evaluation; metric designed for specific tasks; | much diverse input formats; human-aligned, context-specific, open-ended metrics; | fairness; personalization; |

development to the functional analysis of a single function in code. Hence, it calls for better specifications for user requirements/intents, as well as customization to different levels of scope.

## 4.6 Task of Evaluation

In ShareGPT, "evaluation" queries constitute up to 4%. The concept of evaluation permeates almost every aspect of NLP. Standard metrics such as F1 measure (Chinchor, 1992), ROUGE (Lin, 2004), and BERTScore (Zhang et al., 2020) are mostly employed to evaluate classification or generation results at the sentence level. Recently, there is a surge in research efforts to improve alignment with human preferences (Zhong et al., 2022; Liu et al., 2023; Fu et al., 2023; Luo et al., 2023) by using larger models.

However, evaluation-related queries from ShareGPT are quite different. First, we observe that evaluation objects shift from traditional plain texts to a variety of input formats. For instance, GPT is often utilized to assess resumes or code snippets. Furthermore, the evaluation metrics used in these scenarios are exceptionally diverse and open-ended, ranging from the influence of a brand to the feasibility of a plan. This is quite different from previous metrics that are specifically designed for summarization or machine translation.

## 5 Emerging Trends and Challenges

In this section, we summarize the common trends and challenges shared by these prevalent or overlooked tasks we identified from the ShareGPT data.

### 5.1 What trends are reflected in ShareGPT user queries?

In view of user queries in ShareGPT, we notice incredible expansions of task scopes and user bases.

**More Aligned with Daily Life** GPT is leveraged for all kinds of everyday tasks, straddling both professional and personal issues. As such, user queries exhibit an increasing tendency to be free-form and contain arbitrary input, reflecting everyday life scenarios. It is also more customized as a personalized assistant, covering a broad range of topics with nuanced requirements.

**Diverse User Groups** Accompanying the prevalence in everyday tasks is the diversification of user groups. We observe queries by diversifying users of different ages, professions, cultural backgrounds, and even traditionally marginalized groups.

### 5.2 What challenges are proposed by trending and future tasks

Despite the unprecedented success of LLMs, we notice real-world users are also raising their requirements when querying an LLM. Some of the concrete examples are shown in Appendix E.

**Better Reasoning Capacity** One evident need that emerged from user queries is advanced reasoning abilities. Users expect LLMs to comprehend complex scenarios, infer causality, and develop well-organized feasible responses to help them, especially with strategic decision-making.

**Emotion Perceivable** A non-negligible portion of user queries come from marginalized groups seeking help, often due to real-world communication challenges or inadequate solutions. LLMs interacting with these users must effectively perceive their emotions and exhibit empathy, particularly as they may be sensitive when discussing their circumstances, such as those with mental problems. This necessitates a human-centric approach from LLMs, cultivating a safe environment for free expressions

of concerns and offering appropriate support.

**World Knowledge**  In managing the diversity of user queries pertaining to everyday life, the imperative for LLMs to effectively utilize world knowledge grows apparent. This spans from static factual data, such as intercity distances for road trip planning, to dynamic, evolving information like restaurant recommendations fluctuating with Google ratings. Although integrating external plugins (Schick et al., 2023) and applications is a viable initial strategy, meticulous attention must be paid to uphold the neutrality and accuracy of this knowledge, mitigating potential biases and misinformation.

**Multi-modality**  Though restricted to textual interaction with LLMs, user queries demonstrate the potential of a diverse range of modalities. We observe the textual descriptions of images, websites, and UIs, as well as URLs to music/videos in user inputs, which calls for the development of multimodal integrations.

**Personalization and Fairness**  We observe users increasingly expect AI models to understand their unique needs and offer tailored solutions. We also notice the diverse user bases for LLMs of different groups. The drive for personalization must be balanced against the principle of fairness. Personalized responses should not amplify biases or perpetuate unfair outcomes, and the pursuit of fairness should not impede the assistance for customization.

**Dialogue and Interaction**  For user queries that entail massive collaborations or discussions with LLMs, they require a high degree of interactivity, which doesn't just involve passively accepting user queries and providing answers, but actively engaging with users to exchange viewpoints and ideas. This kind of interactive dialogue can help create a more user-friendly experience, facilitating a deeper understanding of the user's needs.

## 6  Conclusion and Future Works

In this paper, we identified a discrepancy between the existing state of NLP research and the need for real-world applications by investigating large collections of ShareGPT and Huggingface data samples with GPT-4. We make this annotation resource public, which could be directly leveraged for further investigation of ShareGPT data, or to fine-tune advanced models such as Llama as a much cheaper

alternative tool for annotation. Based on our observation, we also provide insights into the challenges posed by real user needs and a potential roadmap for future work.

## Limitations

We discuss the limitations of this work in the following aspects:

1. Our study is based on two sources, ShareGPT and Huggingface datasets. Although they are the most abundant resources we can obtain at hand to represent user queries and the traditional benchmarks in the NLP community, they could hardly reflect the whole breadth of real-world situations. Actually, both sets are still growing dynamically as time flows.

2. In our annotation process, we employed GPT-4, the state-of-the-art LLM to help generate domain/topics and task types. On top of that, we conduct a human evaluation for quality assessment. However, there are still chances when the annotation from GPT-4 is not accurate enough, which could influence the post-processing step.

3. Continuing from 2, our work relies on the usage of LLMs. We require annotation from GPT-4 for every sample in ShareGPT and the selected set in Huggingface datasets, which is a large number. Despite that, we make the annotation results publicly available, this annotation process is extremely resource-intensive and time-consuming to reproduce.

## Ethics Statement

Our work highlights the shifted trend and the overlooked problems of previous NLP studies. By analyzing real-world user queries, we examine the new requirements that emerged and hope to make LLMs more beneficial and better aligned with their users' needs, including the marginalized groups. We hope that our work can be an initial effort to mitigate the gap between user needs and academic benchmarks in the era of LLMs. Overall, we do not foresee any major risks or negative societal impacts of our work. The ShareGPT and Huggingface datasets we experiment with are publicly available online. We have open-sourced this project to facilitate future research, especially for small research groups or institutions with relatively fewer resources of LLMs.

## Acknowledgement

Research was supported in part by US DARPA KAIROS Program No. FA8750-19-2-1004, National Science Foundation IIS-19-56151, and the Molecule Maker Lab Institute: An AI Research Institutes program supported by NSF under Award No. 2019897. Any opinions, findings, conclusions, or recommendations expressed herein are those of the authors and do not necessarily represent the views, either expressed or implied, of DARPA, the National Science Foundation, or the U.S. Government.

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

## A Details in Post-processing

In this section, we detail the framework for post-processing. After annotation was completed by GPT-4, we got 1) free-form words and phrases for domain/topic, 2) a one-sentence summary for the user query, and 3) free-form words and phrases task types.

We start by calculating the frequency of certain words/phrases. Considering the existence of synonyms, such as the "advice, tip, suggestion" mentioned before, we resort to an external dictionary[8] and combine synonyms together. On the other hand, those semantically similar words/phrases should also be clustered together. Previously we require GPT-4 to generate summaries for user queries of great quality, and they could be used as references in finding representative samples for the same cluster. For every sample combined by the previous heuristic rules, we search for their k-nearest-neighbors (Keller et al., 1985) and union all the samples as the final results. For words/phrases of very low frequency not incorporated above, we select the nearest "cluster" as the approximation. Finally, to ensure the best possible quality, we manually filtered out unrelated ones in the processed results.

## B Human Evaluation Interface

The website interface screenshot adapted from Doccano for human evaluation is shown in Figure 4. Assessors were informed of the purpose of the study. Before they began to work on the assessment task, they were presented with task instructions shown in Figure 5 and a rating example.

## C Popular programming languages seen in Section 3

We summarize the top 10 mostly used programming languages in coding assistance tasks of ShareGPT. As shown in Figure 3, Python is the mostly used programming language. Apart from that, we observe another large portion of HTML and CSS, reflecting great user needs in website design and programming.

## D Common topics and LLM performance for tasks listed in Section 4

We plot the top 10 mostly seen topics discovered for each task type mentioned in Section 4 shown in Fig-

---

[8] https://www.thesaurus.com/

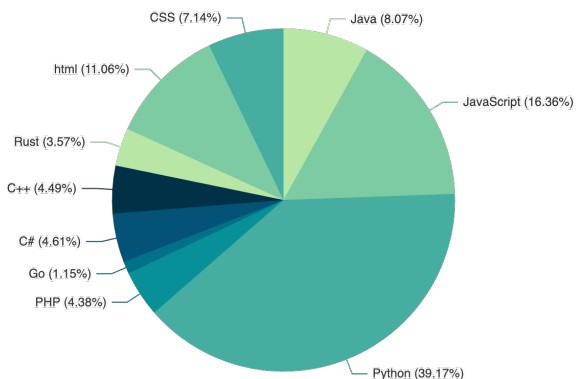

Figure 3: The top 10 most commonly used programming languages in ShareGPT.

ure 6. The shallower color and longer bar indicate a larger portion. For "advice", macro-level guidance such as career advice was the most sought-after category, reflecting individuals' focus on professional growth and job success. This was followed by legal and financial advice, highlighting the everyday complexities people face in navigating legal systems and managing their personal finances. For "design", apart from algorithm design, we also notice creative design requirements such as interface, game, and website, which may also entail visual perception. In "planning", the most common topic discerned was itinerary and travel, indicating significant needs in devising travel plans or daily schedules. As to "discussion", there are explicit topics for discussion such as the historical event or for coding design. We also notice abstract topics like philosophical discussions and ethics discussions, which raises the higher requirement for the self-thinking of LLMs. When it comes to "analysis", a broad range of topics are covered, from specific domains such as literary and historical to personal affairs like relationship analysis. Finally for the task of "evaluation", we notice a diversity of objects to be evaluated, spanning as concrete as a code snippet or grammar and as abstract as candidates for certain positions.

To help better understand the difficulty of the newly identified tasks, we provide a case-based analysis of LLMs. Specifically, we did a preliminary study by randomly selecting 20 samples for each task type from the ShareGPT data. We manually examined the performance of two models on each case, and reported the failure rate for GPT-4 and GPT-3.5-turbo in Table 4 with respect to each task type.

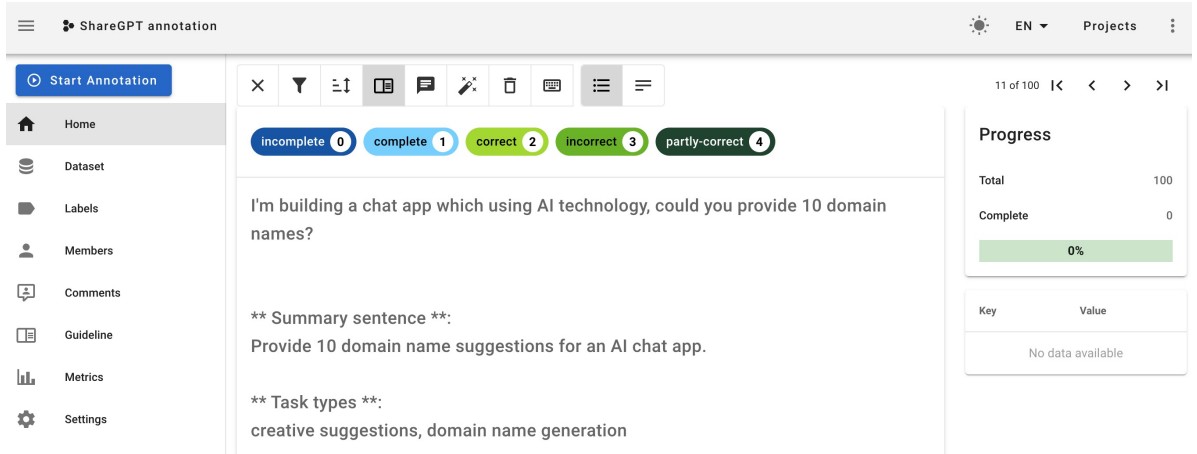

Figure 4: The interface for human assessment. The assessor is shown a user query sampled from ShareGPT, the summary sentence of the user query as a reference, and the generated task types labeled by GPT-4.

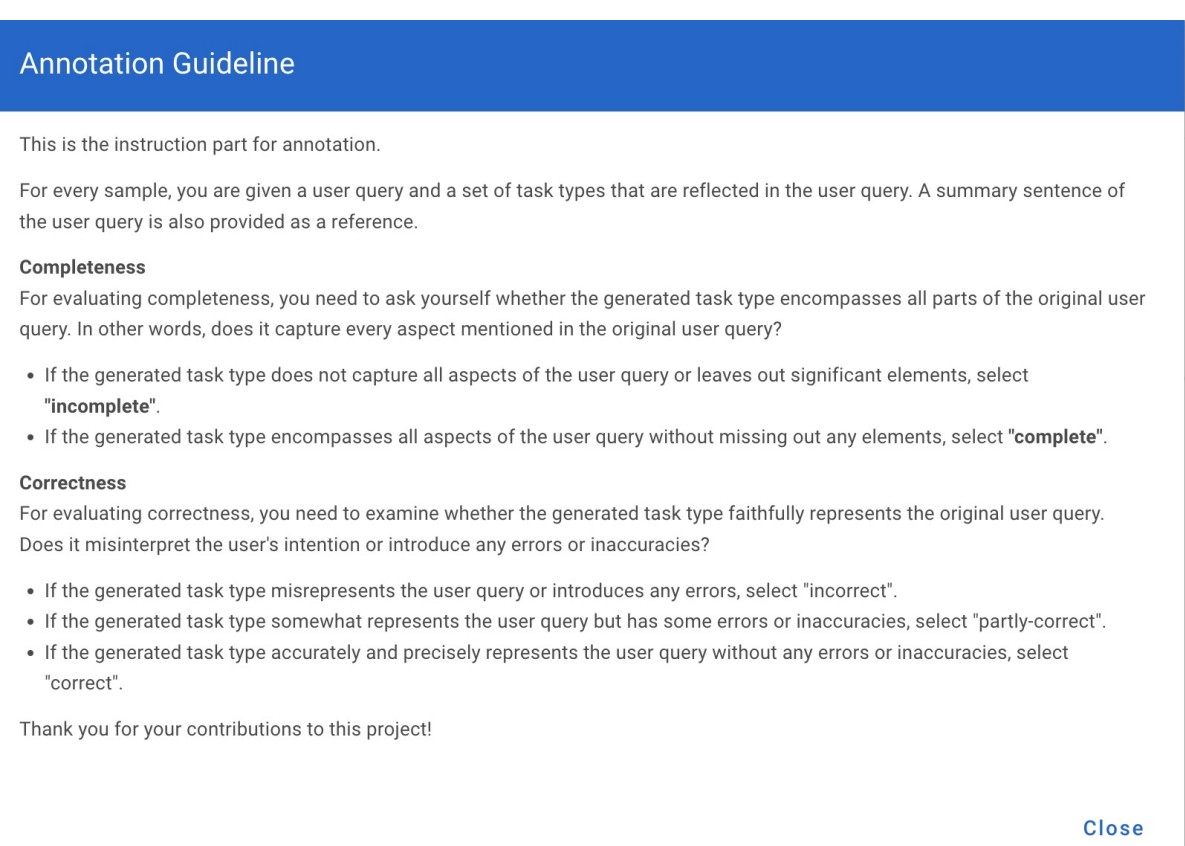

Figure 5: The annotation guideline presented to assessors of our human evaluation process.

## E  Concrete examples mentioned in Section 5

In this part, we display concrete examples in ShareGPT where requirements raised by users pose specific challenges. We highlight the challenging requirements and the misinformation generated by GPT-4 in red.

One example of "advice" about relationship counseling is shown in Figure 10, where the user is seeking emotional support caused by his relationship with his fiancee. We can see that GPT-4 lacks emotional perception, repeating "I'm sorry to hear that..." during the whole interaction, and failing to demonstrate empathy towards the user scenario.

In Figure 7 we display an example of "planning", where the user cast specific constraints on time and places, and specific requirements for EV charging.

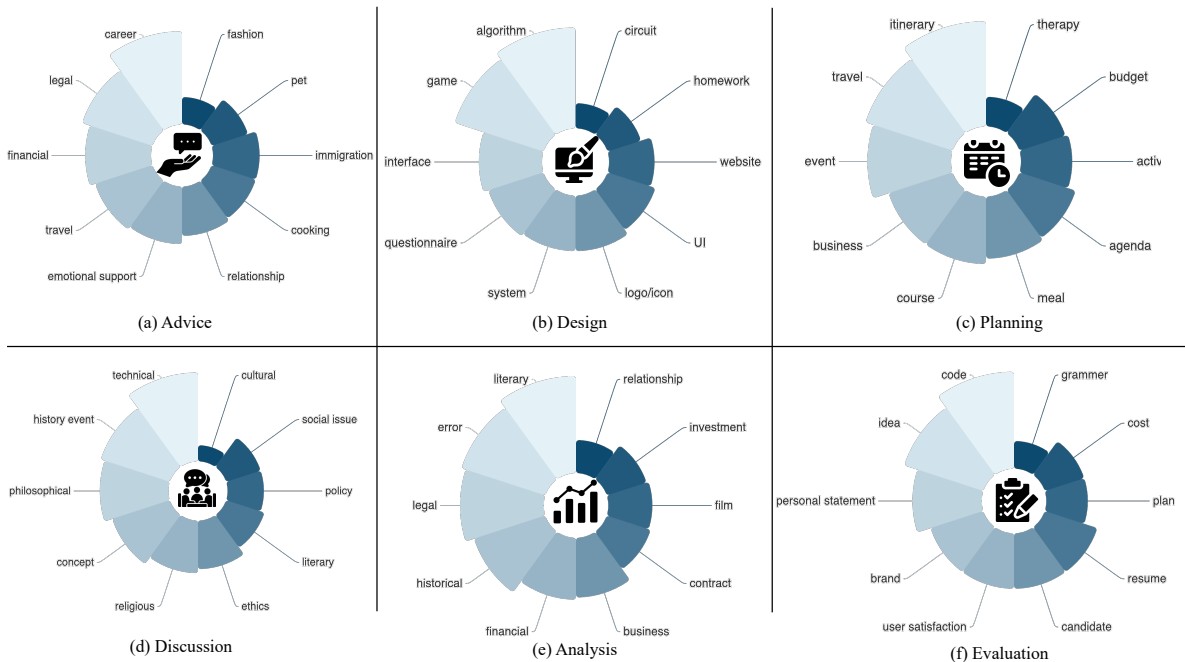

Figure 6: The top 10 most commonly seen topics for the novel tasks discovered from ShareGPT.

Table 4: Case-based analysis for the performance of LLMs.

|  | advice | planning | design | discussion | analysis | evaluation |
|---|---|---|---|---|---|---|
| GPT-3.5-turbo | 0.55 | 0.80 | 0.70 | 0.65 | 0.70 | 0.75 |
| GPT-4 | 0.40 | 0.60 | 0.65 | 0.45 | 0.50 | 0.45 |

GPT-4 generates an 8-day plan, which does not fulfill the requirement for dates. The route is also fully in the UK, failing to cover places like Nice and Calais. Finally, GPT-4 incorporates misinformation for distances between two cities.

In Figure 8 and Figure 9 we visualize two examples related to "design". For course design in Figure 8, the user requires a 5-day design to learn the required content. From GPT-4 response, we can see that the time scheduling is still not feasible to accommodate the content listed, and the design could be overlapped on certain topics. For logo design in Figure 9, GPT-4 does not hold the multimodal ability to deal with this query, and in the conceptual design description, we can see that the design may not fulfill aesthetic beauty requirements and lacks creativity.

In Figure 11 we present another example of "discussion", where the user is initiating a philosophical discussion with GPT. We can see that of the most time, GPT is quoting from existing philosophical viewpoints, and sometimes even demonstrates contradictions in the responses due to the conflict

between user input and the retrieved opinions.

Figure 12 shows an example of "analysis" concerning a character in the play written by Shakespeare. This is a very open and controversial topic for analysis. While in GPT's response, we observe hallucinations, especially towards the change of personality of Bertram, and the misinformation of plots when the character is in Italy.

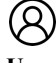

**User**

I'm going on a road trip with my daughter on 3rd April. We will be leaving London and want to be in Nice, France on 9th or 10th April where we'll stay about 3 days. We then need to be in Calais on 15th or 16th to take the Eurotunnel. Everything else is undecided. Can you give a plan for routes where we'll be doing no more than 5 hours driving per day. We have an EV and need to stop for charging.

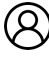

**GPT-4**

Day 1: London to Brighton (2 hrs, 54 miles)
Leave London and head south to Brighton. Brighton is a vibrant seaside town with a great mix of shopping, cultural sites, and delicious food. Visit the iconic Brighton Palace Pier or take a tour of the Royal Pavilion. Make sure to charge your EV overnight.
Day 2: Brighton to Portsmouth (2 hrs, 63 miles)
Drive west along the coast to Portsmouth. Here, you can visit the historic dockyard and Spinnaker Tower for panoramic views of the city. Again, charge your EV overnight.
Day 3: Portsmouth to Poole (2 hrs, 64 miles)
Continue your journey west towards Poole. In Poole, you can visit the lovely Poole Harbour and enjoy the quaint Old Town.
Day 4: Poole to Plymouth (3 hrs, 123 miles)
This is a slightly longer stretch, but Plymouth offers plenty to see, including the National Marine Aquarium and the historic Barbican district.
Day 5: Plymouth to Falmouth (2 hrs, 65 miles)
Head to the picturesque coastal town of Falmouth. Visit Pendennis Castle or take a stroll through the beautiful gardens of Trebah.
Day 6: Falmouth to Bournemouth (3 hrs, 111 miles)
Drive back east to Bournemouth. Here, you can enjoy Bournemouth Beach or visit the Bournemouth Aviation Museum.
Day 7: Bournemouth to Brighton (2 hrs 30 mins, 100 miles)
Return to Brighton, visiting any attractions you may have missed during your first visit.
Day 8: Brighton to London (2 hrs, 54 miles)
Return to London, ready to return to daily life.
The above itinerary involves daily drives of between 2 and 3 hours, leaving you plenty of time to explore each destination. EV charging stations are relatively plentiful in the UK, but you should plan your charging stops in advance and always have a backup plan in case your first-choice charging station is unavailable.

Figure 7: An example of route plan and the corresponding GPT response.

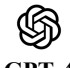

**User**

A deep understanding of the math that makes machine learning algorithms work.\n\n Statistical techniques that empower you to get more out of your data analysis….

Give me a detailed 5 days course design to learn the skills mentioned in the previous question with resources from YouTube books or free coursers and I insist that all the resources are free.

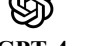

**GPT-4**

Day 1: Fundamentals and Mathematical Background
Morning: 4 hours
1.  MIT OpenCourseWare: Linear Algebra (YouTube): focus on understanding vectors, matrices, singularity, rank, and independence
2.  Khan Academy: Probability and Statistics (YouTube): Brush up your knowledge on the fundamentals of statistics.
Afternoon: 4hours
1.  Andrew Ng's Machine Learning course on Coursera (It's free to audit): only look into sections on Regression (Linear and Logistic)
2.  3Blue1Brown: Essence of linear algebra playlist (YouTube): use this to solidify your understanding of linear algebra concepts
Day 2: Supervised Learning Techniques
Morning: 4 hours
1.  StatQuest with josh Starmer: machine Learning (YouTube) Dive into the supervised learning techniques, focusing on linear and logistic regression.
2.  Sentdex: Practical Machine Learning with Python (YouTube): Get hands-on with examples using Numpy and scikit-learn.
Afternoon: 4 hours
1.  Google's Machine Learning Crash Course: Focus on the sections about neural networks, their architecture, and training.
2.  Sentdex: Neural Networks from Scratch in Python (YouTube): hands-on building and training neural networks.
……

Figure 8: An example of course design and the corresponding GPT response.

| | |
|---|---|
| 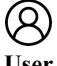 **User** | Design a logo of the company "earth", this company is working in making the world more environment friendly and green, take as much artistic freedom as needed |
| **GPT-4** | As an AI language model developed by OpenAI, I don't have the ability to create visual content such as logos. However, I can provide a description for your logo design based on your instructions. Here is a suggestion:

Logo Design Concept:

The logo should be a stylized representation of the Earth. The Earth can be depicted as a green leaf, symbolizing the company's dedication to environmental sustainability and a greener world.

The leaf could be round like the Earth, with lines indicating the continents subtly visible. This line work should be delicate, indicating the intricate balance and interconnectivity of all parts of the Earth.

The company's name, "earth", could be written in a clean, bold, sans-serif typeface under this symbol to signify the simplicity and clarity of the company's mission. The word "earth" can be in a dark green color to match the overall green theme of the logo.

Consider using shades of green and blue as the primary color palette to further represent the earth's land and sea. A hint of warm yellow or white can be used to represent the sun's energy, which is clean and sustainable.

Please share this description with a graphic designer who can turn these ideas into a visual logo for your company. |

Figure 9: An example of logo design and the corresponding GPT response.

| | |
|---|---|
| **User** | I'm in a hard place. |
| **GPT-4** | I'm sorry to hear that you're going through a difficult time. While I'm an artificial intelligence and don't have personal experiences, I'm here to help provide advice, share useful information, or just listen if you need to talk. Can you tell me a bit more about what you're facing? |
| **User** | My fiancée Madelyn was angry at me tonight. |
| **GPT-4** | I'm sorry to hear that you're having a disagreement with your fiancée, Madelyn. Conflicts in relationships can be stressful. Would you like to talk more about what happened? It might help to sort through your thoughts. Remember, I'm here to help, not to judge. |
| **User** | I have proposed to her; we are engaged. I am committed to her. We have bought a house together. I love her. What else can I do? She was angry at me for the whole night! |
| **GPT-4** | I'm sorry to hear that you're feeling this way. It sounds like you're deeply committed to Madelyn and your relationship with her. It's normal for couples to have disagreements from time to time. Here are a few tips that may help:
**1.Communication**: The first step in any relationship conflict is to talk openly about the problem. Ask her why she was angry. Her reasons might be something you're unaware of, or it might be a misunderstanding. She may also not realize how deeply her anger is affecting you.
**2.Understanding**: Once you understand why she was upset, acknowledge her feelings. Even if you don't necessarily agree with her perspective, understanding where she is coming from is key to resolving the issue. Validating her feelings can show her that you are listening and that you care about her.
...... |
| **User** | If I'm honest, I'm still sad. |
| **GPT-4** | I'm really sorry that you're feeling this way, but I'm unable to provide the help that you need. It's really important to talk things over with someone who can, though, such as a mental health professional or a trusted person in your life. |

Figure 10: An example of relationship counseling and the corresponding GPT response.

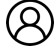
**User**

Ponder this. It cannot be disproven that before my birth and existence, that all time, matter and space and experiences of all, and all opinions, were merely constructed for my arrival in a moment in time, and that all future is also constructed. Including your opinion and all opinions and so-called evidence against the notion, which does not exist, is also constructed. This cannot be disproven.

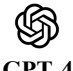
**GPT-4**

The idea you present is a philosophical concept known as solipsism. Solipsism is the philosophical belief that only one's mind is sure to exist. Therefore, everything else, including other people and the external world, may or may not exist. While it cannot be disproven that everything is constructed for your arrival in a moment in time, it also cannot be proven.

The problem with solipsism is that it is unfalsifiable, meaning that it cannot be tested or proven. It is impossible to prove that everything is constructed for your arrival in a moment in time because there is no way to gather evidence for or against this idea. Therefore, it is not a meaningful or useful theory to consider when discussing reality.

Furthermore, the fact that we can have conversations and share ideas suggests that there is a shared reality that exists outside of our individual minds. It is reasonable to assume that there is an objective reality that exists independent of our personal experiences. In conclusion, while solipsism cannot be disproven, it is also not a useful or meaningful concept to consider when discussing reality. It is more reasonable to assume that there is an objective reality that exists independent of our individual experiences.

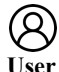
**User**

Firstly, nothing is provable or unprovable. Secondly you said "Furthermore, the fact that we can have conversations and share ideas suggests that there is a shared reality that exists outside of our individual minds". This can be explained by A self and sole existing master fragmented hive mind, that give the rise to the illusion of shared experience, though disconnected smaller minds of an overall master hive fragmented mind.

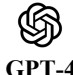
**GPT-4**

While it is true that nothing can be proven or disproven with absolute certainty, in practical terms, we rely on evidence and reasoning to support our beliefs about the world around us. The concept of a self and sole existing master fragmented hive mind is another philosophical idea that cannot be proven or disproven. While it is possible to imagine such a scenario, there is no empirical evidence to support it.

Moreover, the idea of a shared reality goes beyond just the ability to have conversations and share ideas. We can observe and measure the physical world, which suggests that there is an objective reality that exists independently of our individual experiences.

In conclusion, while philosophical concepts like solipsism and the idea of a self and sole existing master fragmented hive mind are interesting to consider, they are not supported by empirical evidence and are not practical ways to understand the world around us. It is more reasonable to assume that there is an objective reality that exists independently of our individual experiences, and that we can rely on evidence and reasoning to support our beliefs about the world.

Figure 11: An example of philosophical discussion with GPT-4.

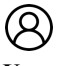
**User**

In the play, All's Well that Ends Well, Bertram is well-known to be one of Shakespeare's most dynamic characters, undergoing a dramatic change from the beginning of the story to the end. Provide a literary analysis of his character arc and explain what makes him so admirable in the end.

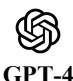
**GPT-4**

"All's Well That Ends Well" is an intriguing play by Shakespeare that straddles the line between comedy and tragedy. The character arc of Bertram is notably complex within the corpus of Shakespearean roles. While some might argue he is a challenging character to admire, the beauty of Bertram's story lies in his journey of growth and understanding.

Bertram, Count of Rousillon, starts the play as a young, ambitious, and somewhat arrogant nobleman who is driven by honor and social status. He is characterized initially by pride, impulsivity, and a certain immaturity, as seen in his refusal to accept Helena, a woman of lower social rank, as his wife despite the king's command and Helena's proven virtue and love for him. He runs away from this marriage and sets seemingly impossible tasks for Helena to fulfill, including obtaining a ring from his finger and bearing his child—believing these to be unachievable.

However, the transformation and character development that Bertram undergoes throughout the play is significant. Over time, he begins to show signs of maturity and understanding. This is mostly seen in the latter part of the play, when he starts to recognize the worth of honesty, loyalty, and love. Helena's unwavering dedication and cunning plan to win him over by accomplishing his tasks, combined with his experiences in the Italian war and interactions with other characters, particularly Diana, help shape Bertram's perception of values and virtues.

The admiration for Bertram in the end does not arise from a newfound perfection in his character, but rather from his human journey from ignorance and pride to understanding and humility. He realizes his mistakes and misjudgments, especially towards Helena, and shows a willingness to make amends. His journey mirrors the human potential for growth and change, giving audiences a sense of hope and faith in the potential for personal redemption.

The phrase that gives the play its title, "All's well that ends well," encapsulates the moral of Bertram's story. Despite his flaws and mistakes, the resolution sees Bertram changed for the better and ready to accept Helena as his wife in spirit as well as in name. His transformation into a more understanding, humble, and compassionate individual provides a sense of satisfactory closure, reinforcing the idea that the outcome of a situation is the most important aspect in evaluating its initial hardship or complexity.

Figure 12: An example of literary analysis of a character with GPT-4.