# OpenReview forum: "The Shifted and The Overlooked: A Task-oriented Investigation of User-GPT Interactions"
_EMNLP/2023/Conference — EMNLP 2023 Main_

### Official Review · Reviewer_XsxS · 2023-08-04

**Soundness:** 3

**Excitement:**

3: Ambivalent: It has merits (e.g., it reports state-of-the-art results, the idea is nice), but there are key weaknesses (e.g., it describes incremental work), and it can significantly benefit from another round of revision. However, I won't object to accepting it if my co-reviewers champion it.

**Paper Topic And Main Contributions:**

Topic: How are real-world queries to LLMs different from traditional NLP benchmarks?

This paper identifies tasks and domains often overlooked in traditional NLP benchmarks by analyzing a collection of user queries to LLMs (ShareGPT). The queries and benchmarks are automatically labeled using GPT-4 to obtain domains, tasks and summaries. They are then clustered and compare to identify shifting trends between the two collections. The paper summarizes the shifting trends and challenges, laying a roadmap to fill the gap between real user needs and academic benchmarks.

**Reasons To Accept:**

1. It investigates a very important problem, i.e. NLP benchmarks should reflect real-world scenarios and queries.
2. It provides a comprehensive analysis of user-GPT conversations and NLP benchmarks.
3. The annotation approach to identify domain, task type and summaries seems reasonable, and aligns with human evaluation.
4. The annotated user-GPT queries is an expensive but valuable resource for future research in NLP.
5. The paper summarizes well the shifting trends and challenges in the era of LLMs.



**Reasons To Reject:**

1. The paper focuses on just two sources, ShareGPT and Huggingface datasets. This may not reflect all the real-world situations. Generally speaking, not all real-world NLP tasks are expected to be solvable by GPT. However, it is important still to have NLP benchmarks for emerging use cases with LLMs.
2. There could also be ethical implications to aligning LLMs with user needs.
3. It would strengthen the paper if the performance of LLMs on the newly identified tasks were also discussed. It will help understand the difficulty of the tasks.
4. There is some overlap in the domains/tasks (which is perhaps difficult to capture simply using embedding similarity).
e.g. relationship vs. personal life; science vs. physics.
The semantics of tasks is also not same across the two sources.
e.g. text generation vs. creative writing. QA is a very broad category (can cover recommendation, IR, comparison etc.)


**Reproducibility:**

3: Could reproduce the results with some difficulty. The settings of parameters are underspecified or subjectively determined; the training/evaluation data are not widely available.

**Reviewer Confidence:**

4: Quite sure. I tried to check the important points carefully. It's unlikely, though conceivable, that I missed something that should affect my ratings.

---

> ### Author Rebuttal · Authors · 2023-08-28
>
> We really appreciate your effort in reviewing our paper and your constructive comments. We are very glad that our research problem is identified as very important, along with comprehensive analysis and valuable annotation resources for future research in NLP. We would like to give detailed responses to each of your comments:
>
> **R1: “The paper focuses on just two sources, ShareGPT and Huggingface datasets. This may not reflect all the real-world situations. Generally speaking, not all real-world NLP tasks are expected to be solvable by GPT. However, it is important still to have NLP benchmarks for emerging use cases with LLMs.”**
>
> A1: Thanks for your comment. We acknowledge the limitation that our analysis is based on the ShareGPT and Huggingface datasets as discussed in the “Limitation” section. Our intent was to provide an investigation into real-world queries and how they differ from traditional benchmarks. Thus providing insights toward making LLMs better aligned with user needs, just as you have noticed. ShareGPT and Huggingface datasets are the most abundant and representative resources we can obtain at hand currently. While we recognize that this doesn't cover every real-world scenario, we believe that these datasets provide a broad representation and that our designed framework and resources could serve as a good starting point for future expanding data sources. For example, the annotation resources could be leveraged to train an advanced model such as Llama for annotating emerging real-world user queries.
>
> **R2: “There could also be ethical implications to aligning LLMs with user needs”.**
>
> A2: We understand the concern about aligning LLMs with user needs. The goal of this paper is to identify gaps and provide insights; (and we are not focusing on specific methods to modify LLMs). However, we'll make sure to modify the existing Ethics Statement and discuss the potential ethical implications in our revised version to provide a more balanced perspective.
>
> **R3: “It would strengthen the paper if the performance of LLMs on the newly identified tasks were also discussed. It will help understand the difficulty of the tasks.”**
>
> A3: Thanks for your insightful comments. A complete qualified evaluation of newly identified tasks is beyond the scope of this paper, and we provide a case-based analysis of model performance in Appendix E with some identified problems.
> Following your suggestion, we did a preliminary study by randomly selecting 20 samples from the ShareGPT data, and reporting the failure rate for GPT-4 and GPT-3.5-turbo in the following table:
>
> |    | advice | planning | design | discussion | analysis | evaluation |
> |  ----  | ----  | ---- |  ----  | ----  | ---- | ---- |
> | GPT-3.5-turbo  | 55% | 80% | 70% | 65% | 70% | 75% |
> | GPT-4 | 40% | 60% | 65% | 45% | 50% | 45% |
>
> **R4: “There is some overlap in the domains/tasks (which is perhaps difficult to capture simply using embedding similarity). e.g. relationship vs. personal life; science vs. physics. The semantics of tasks is also not the same across the two sources. e.g. text generation vs. creative writing. QA is a very broad category (can cover recommendation, IR, comparison etc.)”**
>
> We appreciate your suggestion and we would like to provide further statistics and clarifications.
>
> First, we notice that there are some portions in Figure 2 missing tags due to configurations of Echarts, which potentially lead to this misunderstanding. For domain/tasks overlap, in Figure 2(c), there should be tags of “biology (3.01%)”, “medical (0.81%)” and “chemistry (0.92%)”. In Figure 2(a), the missing tags are “music (3.27%)”, “leisure (2.73%)”, “career (1.35%)”, “food/drink (1.42%)”, “home&living (1.07%)”.
>
> Meanwhile, we decompose “science” in Figure 2(a) into the same scale as “physics” in Huggingface as below:
>
> | neuroscience | chemistry | quantum | biology | cognition&behaviour | geography | medical | agriculture | physics |
> |  ----  | ----  | ---- |  ----  | ----  | ---- | ---- | ---- | ---- |
> | 0.24%  | 0.31% | 0.54% | 0.69% | 0.36% | 0.37% | 1.15% | 0.29% | 1.05% |
>
> Similarly, for “personal life” in Figure 2(c) we get:
>
> | career | home&living | leisure | social media | fashion |
> |  ----  | ----  | ---- |  ----  | ----  |
> | 2.30%  | 1.02% | 1.55% | 1.41% | 0.49% |
>
> For task semantics, the original coarse-grained manner aims to unveil the very basic task type differences such as generation and classification, and highlight that a large portion of existing datasets focus on classification or constrained generation, which is quite different from open-form generation. Following your suggestion, we attached the statistics of major task types in the same scale as below (due to line spacing restrictions, the table is split into two lines of headers):
>
> | text classification | token classification | tabular classification | summarization | fill-mask| table-to-text | IR | code understanding |
> |  ----  | ----  | ---- |  ----  | ----  | ---- | ---- | ---- |
> | 28.13%  | 1.18% | 2.07% | 1.64% | 4.08% | 0.33% | 13.02% | 3.84% |
>
> | reading comprehension | reasoning | IE | analysis | recommendation | translation | planning |
> |  ----  | ----  | ---- |  ----  | ----  | ---- | ---- |
> | 14.23% | 7.76% | 2.30% | 7.76% | 3.06% | 2.52% | 0.53% |
>
> Overall, with the same annotation setting, the domain/topic and task type distributions of huggingface collection are less diverse compare to user queries in ShareGPT. We will revise Figure 2 based on these statistics in the next version of our paper.

---

### Official Review · Reviewer_q1hb · 2023-08-04

**Soundness:** 3

**Excitement:**

3: Ambivalent: It has merits (e.g., it reports state-of-the-art results, the idea is nice), but there are key weaknesses (e.g., it describes incremental work), and it can significantly benefit from another round of revision. However, I won't object to accepting it if my co-reviewers champion it.

**Paper Topic And Main Contributions:**

This paper compares user-GPT conversations with commonly used NLP datasets to identify gaps between the needs of users and the tasks studied by the NLP community. The authors use a snapshot of the ShareGPT dataset to automatically tag each conversation with a domain and task type. They also investigate around 3000 datasets from Huggingface Datasets by sampling 10 instances from each dataset and prompting GPT-4 to classify them. The domains and task types are post-processed in a similar manner for both ShareGPT and Huggingface datasets. The comparison of these GPT-4 tags is used to represent the gaps between user needs and existing NLP datasets.

The main contributions of the paper are:

1. A novel analysis of the kind of interactions users have with GPT compared to the types of problems in NLP datasets. The analysis identifies gaps between the two.

2. A silver-tagged dataset containing the domain and task types of each user conversation in the ShareGPT dataset.

**Reasons To Accept:**

Strengths of the paper:

1. The main strength of the paper is the comparison of the types of questions asked by users to GPT versus the tasks defined in NLP datasets. This comparison is novel and insightful, and it identifies gaps in user demands that the NLP literature can focus on.

2. The paper is very well written. Concrete insights are provided, like trends toward task diversity, open-endedness, and marginalized user groups. Challenges posed by real user needs are summarized, providing a potential roadmap for future work.

**Reasons To Reject:**

Some weaknesses of the paper include:

1. The paper provides a good analysis of the gaps between user needs and NLP datasets, but it does not offer any concrete steps or directions on how these gaps can be addressed. The authors do not discuss what kind of datasets can be generated from what kind of sources, or how the NLP community can actively focus on mitigating the gaps.

2. The dataset provides about 13,700 silver-tagged task types, which makes it difficult to do a concrete analysis of which task types the NLP community should focus on. The authors do not provide any clear criteria for selecting task types, or for prioritizing them.

3. The annotation process relies heavily on GPT-4, which makes reproducing this large-scale annotation resource-intensive and also hard to replicate due to the temperature settings of a specific model.

4. The huggingface datasets currently contains 50k datasets while only 3k datasets were used in the evaluation. The analysis might change quite a bit because only about 6% of Huggingface datasets are being analyzed.

**Reproducibility:**

3: Could reproduce the results with some difficulty. The settings of parameters are underspecified or subjectively determined; the training/evaluation data are not widely available.

**Reviewer Confidence:**

4: Quite sure. I tried to check the important points carefully. It's unlikely, though conceivable, that I missed something that should affect my ratings.

---

> ### Author Rebuttal · Authors · 2023-08-28
>
> We sincerely thank Reviewer q1hb for the positive feedback and valuable suggestions. We appreciate that you recognize that our analysis is novel and that the paper presentation is very well with concrete insights and a potential roadmap summarized. Below we would like to give detailed responses to each of your comments:
>
> **R1: “The paper provides a good analysis of the gaps between user needs and NLP datasets, but it does not offer any concrete steps or directions on how these gaps can be addressed. The authors do not discuss what kind of datasets can be generated from what kind of sources, or how the NLP community can actively focus on mitigating the gaps.”**
>
> Thank you for your comment and for recognizing the value of our gap analysis. Our paper aims to outline some practical challenges and suggest a possible roadmap, as shown in Table 3, for addressing them. We will elaborate more on the following three steps in the revision: 1) data collection, 2) meta-evaluation, and 3) model design.
>
> (i) Data. We do not expect obtaining real user queries to be a major obstacle in the future. Our dataset for analysis comes from ShareGPT, an open-source community that is growing and may release more data. Moreover, products that use LLM, such as Microsoft BingChat, Google Bard, Meta LLaMA, will generate a large number of queries. We can use our technique in this paper to filter the data. Also, as LIMA[1] suggests, we may not need a lot of labeled data to train LLM. Therefore, the dataset collection is feasible with less cost.
>
> (ii) Meta-Evaluation. Evaluating the generation model is a difficult and unsolved problem. GPT-eval is a feasible way for evaluation, as some open-source LLMs, such as Vicuna, do. We could further incorporate human labels/feedback to assess the quality of evaluation metrics and help alleviate potential biases in GPT-eval.
>
> (iii) Model design. We believe this step will draw much attention once we have the training data and the meta-evaluation data. We have many existing models that we can try, such as LLaMA with IR, knowledge, tools, reasoning components, etc.
>
> **R2: “The dataset provides about 13,700 silver-tagged task types, which makes it difficult to do a concrete analysis of which task types the NLP community should focus on. The authors do not provide any clear criteria for selecting task types, or for prioritizing them.”**
>
> A2: Thanks for your suggestion. The 13,700 silver-tagged task types are very fine-grained annotations, spanning “dream analysis” to “markdown formatting”. Our intention is to build a bottom-up approach with abundant information in these fine-grained tags. As mentioned in Section 2.4 and Appendix A, we did post-processing and clustered these task tags into organized task types for analysis.
> Task selection is based on the distribution in Figure 2 that satisfies the following criteria:
>
> (i) The six tasks selected are long-tail tasks of the distribution, with the “body” part in the long-tail phenomenon summing towards around 40%.
>
> (ii) The six tasks are not overly skewed in the distribution, constituting around 25% - a comparable figure to the predominant “code generation” segment.
>
> (iii) These tasks are either overlooked (design, discussion) or exhibit significant shifting features (analysis, advice, planning, evaluation) in NLP literature.
>
> **R3: “The annotation process relies heavily on GPT-4, which makes reproducing this large-scale annotation resource-intensive and also hard to replicate due to the temperature settings of a specific model.”**
>
> A3: Thanks for your comments. We agree that the annotation process is expensive and could be resource-intensive to replicate. We also tried other less expensive or open-source models such as GPT-3.5-turbo and Alpaca/Vicuna, however, the results are far from satisfactory for such a large-scale investigation. We will report the human evaluation results for these models in the next version of our paper. As discussed in “limitation”, we plan to remedy this issue by making the annotation results publicly available. Firstly, this resource could be directly leveraged for further investigation of ShareGPT data. Also, we believe this high-quality annotation source could be used to fine-tune advanced models such as Llama as a much cheaper alternative tool for annotation. We will discuss this in detail in the next version of this work.
>
> **R4: “The huggingface datasets currently contain 50k datasets while only 3k datasets were used in the evaluation. The analysis might change quite a bit because only about 6% of Huggingface datasets are being analyzed.”**
>
> A4: Thanks for your insightful comment. The 3k datasets we used are organized and collected by [2] to ensure diversity and coverage as mentioned in line 286. Please note that although there are 50k datasets in the huggingface data platform, many datasets are not suitable and are thus filtered out in the following four categories:
>
> (i) Datasets with no license description or the license does not allow usage. For ethical considerations, datasets collected are strictly following license restrictions. The number of all datasets fulfilling license requirements are restricted to around 16k.
>
> (ii) Datasets that are non-English. Since our investigation focuses on English language and ShareGPT contains only English samples, datasets collected from huggingface also follow this paradigm.
>
> (iii) Datasets that are related to multi-modal. There is a large portion of datasets in huggingface data platform related to multi-modal research. Since we target at NLP community and related topics, we exclude those datasets.
>
> (iv) Datasets that are invalid. Some datasets in the huggingface data platform are empty or cannot be downloaded, which are ignored by our work.
>
> After these filtering processes, 3k datasets are employed, with 30k instances for annotation in total, which is comparable to the 94k ShareGPT data samples. Please also note that [2] reports performance improvement against baseline models when using this huggingface collection, which also proves the diversity and coverage of this huggingface collection. We will elaborate more on this point in the next version of our paper.
>
> [1] Zhou, Chunting, et al. "Lima: Less is more for alignment." arXiv preprint arXiv:2305.11206 (2023).
>
> [2] Yin, Da, et al. "Dynosaur: A Dynamic Growth Paradigm for Instruction-Tuning Data Curation." arXiv preprint arXiv:2305.14327 (2023).

---

### Official Review · Reviewer_XWHq · 2023-08-07

**Soundness:** 5

**Excitement:**

5: Transformative: This paper is likely to change its subfield or computational linguistics broadly. It should be considered for a best paper award. This paper changes the current understanding of some phenomenon, shows a widely held practice to be erroneous in someway, enables a promising direction of research for a (broad or narrow) topic, or creates an exciting new technique.

**Paper Topic And Main Contributions:**

The central objective of this paper is to identify the diverging gap between the current NLP research and the need of real-world NLP applications. To do this, this work uses a data set collection of user-chatbot conversations collected via the ShareGPT application. The authors use GPT4 for processing this collected data to obtain the topic/domain, task type and summary. Following this they carry out a series of experiments to learn about the real-world user query distribution and what are the emerging tasks and requirements that are missing from current NLP research/benchmarks.

**Reasons To Accept:**

* The paper is well-written in an easy-to-understand manner. All the experimental design and setups are clearly explained. The annotation and evaluation strategies are also well-designed.

* This paper discusses an important topic for NLP research community, i.e., the discrepancy between the existing state of NLP research and the needed NLP directions from real-world applications.

   * For this, they use a large enough collection of user chats ~94k samples.

   * For annotation of the task type and domain, they develop a neat and simple pipeline using GPT4 with chain-of-thought prompting and in-context prompting methods. This results in ~13.7k task types and ~8.3k domain labels. Human evaluations carried out for correctness and completeness have strong scores and kappa for each validating GPT4's use for annotation.

   * After post-processing and matching the domain and type distributions with 2.9k datasets available on huggingface, they deliver their insights systematically by discussing the long tail distributions of the tasks discovered, the various shifts and the unnoticed research gaps they learn from their study. They have done this with proper quantifying experiments and by proving supporting examples and references to relevant works of datasets and evaluation when necessary in their write-up.

* In my opinion, the insights brought out by this work are clearly of high value to the NLP community at the present time.

**Reasons To Reject:**

* No strong reasons to reject

**Reproducibility:**

4: Could mostly reproduce the results, but there may be some variation because of sample variance or minor variations in their interpretation of the protocol or method.

**Reviewer Confidence:**

4: Quite sure. I tried to check the important points carefully. It's unlikely, though conceivable, that I missed something that should affect my ratings.

---

> ### Author Rebuttal · Authors · 2023-08-28
>
> We sincerely thank Reviewer XWHq for the strong review and the time you spent on our submission. We really appreciate your recognition of our commendable paper presentation, experimental design, and evaluation. We are also very glad that you think our research topic is important and of high value to the current NLP community. We will continue to polish our work and thanks for arguing in our favor!

---

### Meta-Review · Area_Chair_Nz9C · 2023-09-14

**Recommendation:** 5

**Metareview:**

The main conclusions of the reviews and the post-rebuttal discussions:

3/ 3 reviewers consider the paper sound (scores 5, 3, 3)
3/ 3 reviewers find the paper exciting (scores 5, 3, 3)

From reading the rebuttal and seeing the scores above, I find that the reviewers consider strong points for soundness the following:
- This paper discusses an important topic for NLP research community, i.e., the discrepancy between the existing state of NLP research and the needed NLP directions from real-world applications
- Well-designed and well-explained experimental and evaluation setup
- The comparison of the types of questions asked by users to GPT versus the tasks defined in NLP datasets is novel and insightful, and it identifies gaps in user demands that the NLP literature can focus on.
- Concrete insights are provided, like trends toward task diversity, open-endedness, and marginalized user groups. Challenges posed by real user needs are summarized, providing a potential roadmap for future work

However, there are a few concerns expressed by the reviewers regarding the ethical implications to aligning LLMs with user needs, the difficulty of replicating the annotations given the reliance on GPT-4, and the focus on just two datasets might not reflect all real-world situations.
The reviewers respond to these concerns and plan to address them in the limitation section of their paper.

---

### Decision · Program_Chairs · 2023-10-07

**Decision:**

Accept-Main

**Comment:**

The main conclusions of the reviews and the post-rebuttal discussions:

3/ 3 reviewers consider the paper sound (scores 5, 3, 3)
3/ 3 reviewers find the paper exciting (scores 5, 3, 3)

From reading the rebuttal and seeing the scores above, I find that the reviewers consider strong points for soundness the following:
- This paper discusses an important topic for NLP research community, i.e., the discrepancy between the existing state of NLP research and the needed NLP directions from real-world applications
- Well-designed and well-explained experimental and evaluation setup
- The comparison of the types of questions asked by users to GPT versus the tasks defined in NLP datasets is novel and insightful, and it identifies gaps in user demands that the NLP literature can focus on.
- Concrete insights are provided, like trends toward task diversity, open-endedness, and marginalized user groups. Challenges posed by real user needs are summarized, providing a potential roadmap for future work

However, there are a few concerns expressed by the reviewers regarding the ethical implications to aligning LLMs with user needs, the difficulty of replicating the annotations given the reliance on GPT-4, and the focus on just two datasets might not reflect all real-world situations.
The reviewers respond to these concerns and plan to address them in the limitation section of their paper.